# Frequency-Enhanced Hybrid Multimodal CNN-Transformer Network for Electrocardiogram Classification

1st Yufeng Wei
*School of Automation*
*Wuhan University of Technology*
Wuhan, China
yufengwei@whut.edu.cn

2nd Cheng Lian
*School of Automation*
*Wuhan University of Technology*
Wuhan, China
chenglian@whut.edu.cn

*Abstract*—Recently, deep learning-based models have been widely adopted for electrocardiogram (ECG) classification tasks, demonstrating greater accuracy and efficiency than manual diagnosis. Most existing methods use raw ECG or its time-frequency domain representation as input. These methods are constrained by their reliance on a single input modality, thereby limiting the network's ability to capture discriminative information effectively. In our study, we treat the frequency spectrum of ECG as an independent modality and input it into a multimodal classification model along with ECG. Our method combines depthwise separable convolution and Transformer architectures for unimodal feature extraction. A linear layer aligns the features from both modalities, and a Transformer layer facilitates multimodal feature fusion. We evaluate the performance of our model in both the multi-label classification task using the Ningbo dataset and the multi-class classification task using the Real World dataset. Our model demonstrates superior classification performance compared to the competitive baseline models.

*Index Terms*—ecg, frequency spectrum, multimodal fusion

## I. INTRODUCTION

Cardiovascular disease stands as one of the primary causes of mortality worldwide [1]. Early detection of cardiovascular disease is crucial to initiate effective treatment. ECG reflects the electrical activity of heart and is widely used to monitor heart health. Initially, the analysis of ECG relied on human experts. However, this manual diagnosis proves not only time-intensive and laborious but also susceptible to subjectivity and variability in professionalism [2]. To address the limitations inherent in manual diagnosis, several studies have explored the application of traditional machine learning techniques, such as support vector machines (SVM) and random forests (RF), in classifying manually extracted features from ECG [3, 4]. While these methods offer good interpretability, they fail to accurately capture the diverse variations in ECG across different environments and populations [5].

In recent years, deep learning models have been widely used in ECG classification tasks. These models leverage data-driven techniques to dynamically extract discriminative features, resulting in excellent classification performance. For instance, Xu *et al.* [6] introduce a model based on a coupled convolutional layer structure for heartbeat classification, achieving an overall accuracy of 99.43%. Similarly, Sun *et al.* [7] propose a recurrent neural network (RNN) comprising stacked long short-term memory (LSTM) units for atrial fibrillation (AF) prediction, yielding a F1 score of 92%. Moreover, in [8], an 8-layer convolutional neural network (CNN) with shortcut connections is combined with a 1-layer long short-term memory (LSTM) network to enhance the model's capacity to capture long-term dependencies. However, these methods focus on leveraging time domain information and often overlook the wealth of knowledge embedded in the frequency domain of ECG.

Prior research has underscored the efficacy of incorporating frequency domain information into ECG classification tasks [9–11]. These studies either utilize time-frequency spectrogram as input or employ encoders to represent information in both time and frequency domains. While the former method remains confined to modeling a single modality, the latter method, by separately modeling the frequency domain, offers a more comprehensive exploration of valuable information inherent in frequency domain.

In this paper, we treat the frequency spectrum of ECG as an independent modality and propose a multimodal feature extraction and fusion model for ECG classification. Specifically, the frequency spectrum is generated via fast Fourier transform (FFT) and subsequently fed into an encoder separately from the ECG. We devise a CNN layer mainly comprising depthwise separable convolution to initially extract unimodal features, followed by the utilization of a Transformer layer to capture long-term dependencies. To facilitate multimodal fusion more effectively, a linear layer is employed to align the features of the two modalities. Subsequently, another Transformer layer is utilized for the final feature fusion. Our main contributions can be summarized as follows:

- Proposal of a multimodal model encompassing feature extraction, alignment, and fusion processes tailored for ECG classification.
- Development of a feature extraction module comprising depthwise separable convolution.
- Utilization of Transformer architecture to model long-

term dependencies of features and facilitate multimodal feature fusion.

The subsequent sections of this paper are organized as follows: Section II describes the related work concerning deep learning models for ECG classification. Section III describes the proposed method in detail. Section IV describes the experimental design and presents the analysis of the experimental results. Section V describes the conclusions drawn from this study and outlines potential directions for future research.

## II. RELATED WORK

### A. Deep Learning Models for ECG Classification

Most ECG classification models based on deep learning leverage CNNs due to their robust feature extraction capabilities. For example, Niu *et al.* [12] develop a multi-view convolutional neural network (MPCNN) that employs symbolic representations of heartbeats to automatically learn features and classify heartbeats. Additionally, Wang [13] propose an automatic AF detection method using an 11-layer neural network, primarily composed of a CNN and an improved Elman neural network (MENN). Some researchers also explore the use of LSTM or bidirectional LSTM for ECG classification, which mainly considers the temporal properties of ECG [14, 15]. To construct more powerful models, LSTM and attention mechanism are introduced and combined with CNN, further enhancing classification performance [8, 16]. Furthermore, the vision transformer, a popular model in the field of computer vision, is introduced into ECG classification tasks, typically involving the transformation of ECG into a two-dimensional image [10].

## III. METHODS

This section provides a detailed description of our method. As illustrated in Fig. 1 (a), frequency spectrum is generated from ECG via the FFT and then fed into our model along with ECG. The extraction of multimodal features involves two stages: unimodal feature extraction and multimodal feature fusion. In the first stage, a carefully designed CNN layer and a Transformer layer are employed for unimodal feature extraction. In the second stage, a linear layer is used for feature alignment, and a Transformer layer facilitates multimodal feature fusion. The final multimodal features are then subjected to global average pooling before being fed into a linear layer for classification. The specifics of the network architecture are detailed in Table I, which represents the Base version. We also provide a Small version of our model, in which the number of CNN blocks is reduced to 2, the number of downsampling layers is reduced to 1, and the dimension of feature channels is halved. The following sections will introduce the different components in detail.

### A. Frequency Spectrum Generation

Given an ECG $\mathbf{s}^{\text{t}} \in \mathbb{R}^{N \times L}$, where $N$ is the number of leads and $L$ is the length of each lead, the frequency spectrum

## TABLE I
DETAILED ARCHITECTURE OF OUR MODEL

| Module | Layer Details |
|---|---|
| Value Embedding | $in\_channel = lead\_num; out\_channel = 32$ 
 $kernel\_size = 3; stride = 1; padding = 1$ |
| CNN block$\times 3$ | $in\_channel = 32, 64, 128; out\_channel = 32, 64, 128$ 
 $DC : kernel\_size = 9; stride = 1; padding = 4$ 
 $PC : kernel\_size = 1; stride = 1; padding = 0$ |
| Downsample $\times 2$ | $in\_channel = 32, 64; out\_channel = 64, 128$ 
 $kernel\_size = 2, 2; stride = 2, 2; padding = 0, 0$ |
| Transformer block $\times 2$ | $embed\_dim = 128, num\_head = 2$ |

$\mathbf{s}^{\text{f}} \in \mathbb{R}^{N \times L}$ is obtained using the FFT. The transformation formula for $i$ th lead is as follows:

$$\mathbf{s}_i^{\text{f}}(k) = \text{ABS}(\text{FFT}[\mathbf{s}_i^{\text{t}}]) = \text{ABS}(\sum_{l=0}^{L-1} \mathbf{s}_i^{\text{t}}(l) \cdot e^{-j\frac{2\pi}{L}lk},$$
$$l = 0, 1, \ldots, L-1 \quad (1)$$

where $k$ represents the different frequencies in ECG, and ABS denotes the operation of taking the absolute value, which ensures that only the amplitude of frequency spectrum is retained. The transformed frequency spectrum has the same length as the corresponding ECG but does not belong to time series. Instead, it reflects the essential characteristics of ECG from a different perspective.

### B. Unimodal Feature Extraction

This part of the network comprises three components: the Value Embedding layer, the CNN layer, and the Transformer layer.

1) **The Value Embedding layer**: This layer consists of a 1D convolutional layer followed by a batch normalization (BN) layer to adjust the channel dimensions and initially capture the local relationships within the data. The encoding process for modality m is described by the following formula:

$$\mathbf{x}_{\text{VE}}^m = \text{BN}(\text{Conv1D}(\mathbf{s}^m)) \quad (2)$$

where $m$ represents ECG or frequency spectrum.

2) **The CNN layer**: Our design of the CNN layer refers to ConvNeXt [17]. It comprises stacked CNN blocks with downsampling layers interspersed between different blocks. The structure of a single CNN block is depicted in Fig. 1 (b). Depthwise convolution (DC) is employed to model the feature of each channel independently, followed by point convolution (PC) to capture inter-channel interactions. The PC operation is structured to expand and then contract the channel dimension, utilizing two different point convolutions. A BN layer or a GELU activation function is inserted between different convolution layers, and a residual connection is applied to the final output. The downsampling layer consists of a BN layer and a 1D convolutional layer, which doubles the channel dimension of the features while reducing its length by half.

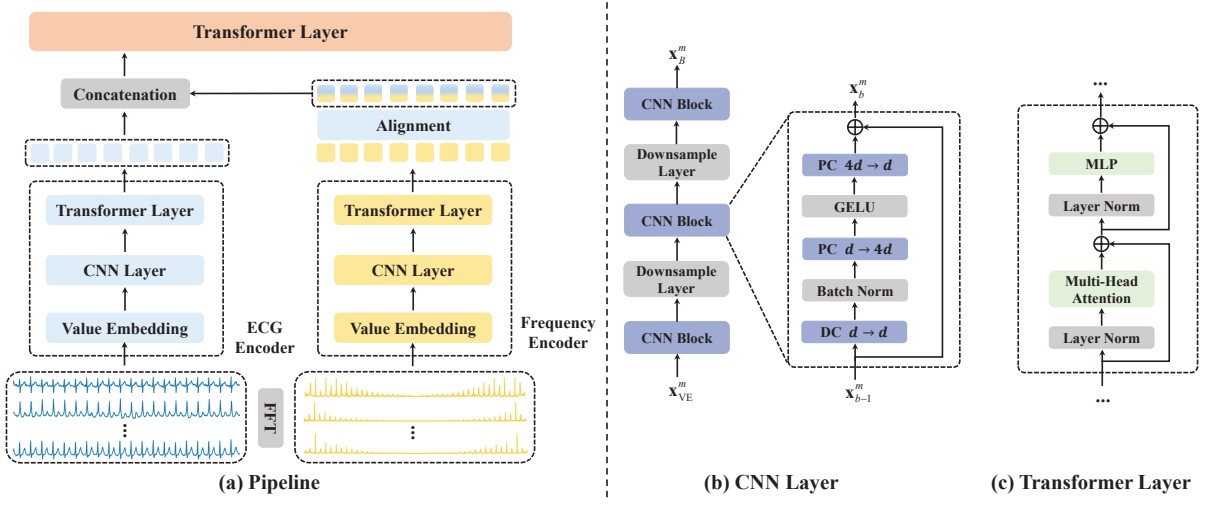

**Fig. 1.** The pipline and various components in our model.

The output of the $b$ th CNN Block can be described by the following formula:

$$\tilde{\mathbf{x}}_b^m = \mathrm{BN}(\mathrm{DC}_{C \to C}(\mathbf{x}_{b-1}^m)) \tag{3}$$

$$\hat{\mathbf{x}}_b^m = \mathrm{PC}_{4C \to C}(\mathrm{GELU}(\mathrm{PC}_{C \to 4C}(\tilde{\mathbf{x}}_{b-1}^m))) + \mathbf{x}_{b-1}^m \tag{4}$$

$$\mathbf{x}_b^m = \mathrm{DownSample}(\hat{\mathbf{x}}_b^m) = \mathrm{BN}(\mathrm{Conv1D}(\hat{\mathbf{x}}_b^m)) \tag{5}$$

where $C$ represents the channel dimension of the features.

3) **The Transformer layer**: To capture long-term dependencies within the features, we utilize a Transformer block as depicted in Fig. 1 (c). The Transformer block consists of a multi-head self-attention layer (MSA) and a multi-layer perceptron (MLP) layer. Layer normalization (LN) layers and residual connections are applied before and after each component. The output of the CNN layer is represented as $\mathbf{x}_B^m$, and the output of the Transformer layer can be described by the following formula:

$$\tilde{\mathbf{x}}_\mathrm{T}^m = \mathrm{MSA}(\mathrm{LN}(\mathbf{x}_B^m)) + \mathbf{x}_B^m \tag{6}$$

$$\mathbf{x}_\mathrm{T}^m = \mathrm{MLP}(\mathrm{LN}(\tilde{\mathbf{x}}_\mathrm{T}^m)) + \tilde{\mathbf{x}}_\mathrm{T}^m \tag{7}$$

where $\mathbf{x}_\mathrm{T}^m$ represents the output of the Transformer layer.

*C. Multimodal Feature Fusion*

After obtaining the features from the ECG and frequency spectrum $\mathbf{x}_\mathrm{T}^\mathrm{t}$ and $\mathbf{x}_\mathrm{T}^\mathrm{f}$, we use a linear layer to align the features of the two modalities. This method has been effectively applied to image-text alignment [18, 19]. Specifically, the linear layer is applied only to the features of the frequency spectrum to align it with the features of the ECG. Subsequently, the features from both modalities are concatenated and fed into a Transformer block for final fusion. The process is described by the following formula:

$$\tilde{\mathbf{x}}_\mathrm{T}^\mathrm{f} = \mathrm{Linear}(\mathbf{x}_\mathrm{T}^\mathrm{f}) \tag{8}$$

$$\mathbf{x}_\mathrm{out} = \mathrm{Transformer}([\mathbf{x}_\mathrm{T}^\mathrm{t}; \tilde{\mathbf{x}}_\mathrm{T}^\mathrm{f}]) \tag{9}$$

where $\mathbf{x}_\mathrm{out}$ represents the output of the Transformer layer, $[;]$ represents concatenation operator.

## IV. EXPERIMENT

*A. Datasets*

To verify the effectiveness of our proposed model, we conduct experiments on two ECG datasets: the Ningbo dataset [20] from the Computing in Cardiology Challenge (CinC) 2021 database [21] and the Real World dataset which is collected by ourselves. The Ningbo dataset comprises 34,905 samples with a sampling frequency of 500 Hz and includes 25 classes representing different heart rhythm types. Each sample is 10 seconds long and contains multiple labels. The Real World dataset consists of 1,091 samples, with a sampling frequency of 1,000 Hz and 3 classes (normal, AF, and others). The duration of each sample in the Real World dataset is variable, and each sample has a single label. For consistency in processing, we extract a 10-second segment from each sample for classification. We utilize stratified sampling to partition 80% of each dataset into a training set and 20% into a test set.

*B. Metrics*

The two datasets represent multi-label and multi-class classification tasks, respectively. For the multi-label classification task, we use the following evaluation metrics: accuracy (Acc), sample F1 score (Sample-F1), area under the receiver operating characteristic curve (AUROC), and area under the precision-recall curve (AUPRC). For the multi-class classification task, we use accuracy (Acc), macro F1 score (Macro-F1), precision (Pre), and recall (Rec) as evaluation indicators.

*C. Compared Methods*

We compare our method with several popular baselines.
1) **LSTM** [22] : Long short-term memory networks (a special type of RNN).
2) **BiLSTM** [23] : Bidirectional LSTM networks can better capture long-term dependencies in both directions.

TABLE II
COMPARISON AMONG OUR MODEL AND BASELINE METHODS.

| Models | Ningbo | | | | Real World | | | |
|---|---|---|---|---|---|---|---|---|
| | Acc | Sample-F1 | AUROC | AUPRC | Acc | Pre | Rec | Macro-F1 |
| LSTM | 53.22 | 73.08 | 90.76 | 40.46 | 68.85 | 53.16 | 57.52 | 53.08 |
| BiLSTM | 51.57 | 74.80 | 89.51 | 43.99 | 67.14 | 54.50 | 53.60 | 52.23 |
| ViT | 47.51 | 65.65 | 90.54 | 41.34 | 68.21 | 62.75 | 52.98 | 55.19 |
| MobileNetV3 | 63.63 | 83.55 | 94.09 | 55.87 | 77.66 | 61.16 | 59.57 | 59.85 |
| Xresnet1d101 | 60.46 | 81.04 | 95.40 | 55.96 | 82.53 | 65.22 | 69.97 | 66.30 |
| ISIBrno-AIMT | 51.21 | 75.10 | 91.33 | 45.23 | 84.70 | 68.86 | 65.20 | 63.05 |
| Ours | **64.02** | **83.56** | **95.62** | **60.87** | **88.77** | **79.81** | **70.44** | **70.95** |

TABLE III
ABLATIONS OF OUR MODEL.

| Models | Ningbo | | | | Real World | | | |
|---|---|---|---|---|---|---|---|---|
| | Acc | Sample-F1 | AUROC | AUPRC | Acc | Pre | Rec | Macro-F1 |
| ECG | 58.54 | 78.76 | 93.61 | 56.37 | 86.32 | 70.66 | 70.12 | 69.53 |
| Frequency Spectrum | 56.58 | 76.84 | 88.70 | 39.70 | 72.31 | 53.35 | 58.97 | 52.73 |
| w/o long dependency | 62.57 | 83.05 | 93.93 | 56.99 | 88.93 | 66.92 | 60.08 | 60.75 |
| w/o alignment | 62.40 | 83.04 | 95.34 | 60.03 | 80.76 | 68.82 | 72.70 | 68.54 |
| Ours | **64.02** | **83.56** | **95.62** | **60.87** | **88.77** | **79.81** | **70.44** | **70.95** |

3) **ViT** [24] : The Vision Transformer uses patches as the smallest processing unit and relies on the Transformer architecture for modeling.

4) **MobileNetV3** [25] : A lightweight CNN architecture suitable for mobile devices.

5) **Xresnet1d101** [26] : A ResNet network for time series.

6) **ISIBrno-AIMT** [16] : It combines a CNN and a attention mechanism for heart rhythm classification.

### D. Implementation Details

Considering the size of the two datasets, we use the Base version of the model for the Ningbo dataset and the Small version for the Real World dataset. For a fair comparison, the training parameters for all networks are kept consistent in the experiments: the batch size is set to 128, the learning rate is set to 0.001, the Adam optimizer is used, and the number of training epochs is set to 100. All experiments are performed on an NVIDIA 2080Ti GPU.

### E. Experimental Results Analysis

Table II presents the experimental results on the two datasets. All indicators are expressed as percentages (%), with the best results highlighted in **bold**. Overall, our model demonstrates superior performance on both datasets. On the Ningbo dataset, the AUPRC is improved by 9% compared to the suboptimal MobileNetV3. On the Real World dataset, the macro F1 score is improved by 7% compared to the suboptimal Xresnet1d101. We attribute this phenomenon to the effective combination of CNN and Transformer architectures and the introduction of multimodal technology. Additionally, we observe that CNN-based models (MobileNetV3 and Xresnet1d101) often outperform LSTM-based and Transformer-based models, indicating that CNNs can extract more discriminative features

for ECG classification tasks. The combined model, which leverages both CNN and Transformer architectures, harnesses the strengths of both architectures and achieves superior classification performance through their effective integration.

### F. Ablation Study

We conduct ablation experiments on the model and explored it from three perspectives: 1) using only a single modality as input (ECG or Frequency Spectrum), 2) omitting the use of the Transformer layer to capture long-term dependencies when extracting unimodal features (w/o long dependency), and 3) not using the linear layer for modality feature alignment (w/o alignment). Table III shows the performance of the model in these three cases. The results indicate that when only ECG or frequency spectrum is used as input, the classification performance of the model is lower than that of the full model, demonstrating that the introduction of multimodal technology enhances the model's discriminative ability. In the scenarios of w/o long dependency and w/o alignment, the classification performance of the model decreases, further proving the effectiveness of these components.

## V. CONCLUSION

In this paper, we propose a multimodal model combined with frequency spectrum for ECG classification. Depthwise separable convolution is employed to initially extract unimodal features, and a Transformer layer is used to capture long-term dependencies. Our method incorporates a linear layer for modality feature alignment and a Transformer layer for multimodal feature fusion. Our model outperforms competitive baselines in both multi-label and multi-class classification tasks, demonstrating its effectiveness. In the future, we aim to design classification models using more modalities for

heart disease diagnosis and integrate exogenous data to further enhance the model's performance.

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
