# OpenReview forum: "Frequency-Enhanced Hybrid Multimodal CNN-Transformer Network for Electrocardiogram Classification"
_IEEE.org/ICIST/2024/Conference — IEEE ICIST 2024 Conference Submission_

### Official Review · Reviewer_SUuR · 2024-08-25
**A frequency-enhanced hybrid multimodal CNN-transformer network is proposed for the electrocardiogram classification. It is interesting and meaningful.**

**Rating:** 7
**Confidence:** 3

**Review:**

1. The contributions of this paper are listed in the blow of Section I. However, the descriptions of the achieved effect is insufficient.

2. Some annotations of variables in formulas are missing. Please replenish them.

---

### Official Review · Reviewer_3Bf3 · 2024-09-03
**Frequency-Enhanced Hybrid Multimodal CNN-Transformer Network for Electrocardiogram Classification**

**Rating:** 7
**Confidence:** 4

**Review:**

The paper is well-structured and easy to follow. The introduction clearly outlines the motivation and limitations of existing ECG classification methods. However, some minor improvements in clarity could further enhance readability. For example, the details of the proposed method in Section III could benefit from additional subheadings to differentiate between the different components (e.g., Frequency Spectrum Generation, Unimodal Feature Extraction, and Multimodal Feature Fusion). This would make it easier for readers to navigate and understand the technical aspects of the method.

---

### Official Review · Reviewer_a99L · 2024-09-03
**This paper can be considered for publication.**

**Rating:** 7
**Confidence:** 2

**Review:**

The authors in this paper investigate the frequency-enhanced hybrid multimodal CNN-transformer network for electrocardiogram classification. The reviewer has the following comments.
1. Would the experiment results be changed with the change of testing datasets?
2. Some typos and grammar errors are found in the paper. The authors are suggested to carefully double-check the manuscript.

---

### Decision · Program_Chairs · 2024-09-06

Accept (Oral)